# Three-Dimensional Morphometric Analysis of the Volar Cortical Shape of the Lunate Facet of the Distal Radius

**DOI:** 10.3390/diagnostics14161802

**Published:** 2024-08-18

**Authors:** Yusuke Eda, Reo Asai, Sho Kohyama, Akira Ikumi, Yasukazu Totoki, Yuichi Yoshii

**Affiliations:** 1Department of Orthopaedic Surgery, Tsukuba Medical Center Hospital, 1-3-1, Amakubo, Tsukuba 305-8576, Japan; 2Department of Orthopedic Surgery, Tokyo Medical University Ibaraki Medical Center, Ami 300-0395, Japan; reo.a@tsukuba-seikei.jp; 3Department of Orthopedic Surgery, Kikkoman General Hospital, Noda 278-0005, Japan; 4Department of Orthopedic Surgery, Faculty of Medicine, University of Tsukuba, Tsukuba 305-8577, Japanyasutotoki@tsukuba-seikei.jp (Y.T.)

**Keywords:** distal radius fracture, lunate facet, computer analysis, volar locking plate

## Abstract

In cases of distal radius fractures, the fixation of the volar lunate facet fragment is crucial for preventing volar subluxation of the carpal bones. This study aims to clarify the sex differences in the volar morphology of the lunate facet of the distal radius and its relationship with the transverse diameter of the distal radius. Sixty-four CT scans of healthy wrists (30 males and 34 females) were evaluated. Three-dimensional (3D) images of the distal radius were reconstructed from the CT data. We defined reference point 1 as the starting point of the inclination toward the distal volar edge, reference point 2 as the volar edge of the joint on the bone axis, and reference point 3 as the volar edge of the distal radius lunate facet. From the 3D coordinates of reference points 1 to 3, the bone axis distance, volar−dorsal distance, radial−ulnar distance, 3D straight-line distance, and inclination angle were measured. The transverse diameter of the radius was measured, and its correlations with the parameters were evaluated. It was found that in males, compared to females, the transverse diameter of the radius is larger and the protrusion of the volar lunate facet is greater. This suggests that the inclination of the volar surface is steeper in males and that the volar locking plate may not fit properly with the volar cortical bone of the lunate facet, necessitating additional fixation.

## 1. Introduction

The morphology of the distal radius is of great interest in orthopedic trauma surgery due to its significant variability across individuals, including differences by gender and age [1,2]. Traditionally, morphological indicators such as radial inclination and volar tilt have been assessed using two-dimensional (2D) imaging modalities, primarily plain radiographs (X-rays) [3,4]. These measurements have provided valuable information about the structural integrity and alignment of the distal radius. However, the advent of three-dimensional (3D) computed tomography (CT) has allowed for more detailed and localized pathophysiological evaluations [5,6,7].

Unlike traditional X-rays, 3DCT offers a comprehensive view of bone anatomy, enabling precise measurements that capture the complex geometry of the distal radius. In previous studies, 3D morphological measurements of the distal radius using 3DCT have been conducted, enabling localized morphological evaluations and measurements [8,9,10,11]. This transition has been particularly beneficial in the context of distal radius fractures, for which accurate assessment of bone morphology is crucial for effective treatment planning and execution.

The widespread adoption of volar locking plates has significantly improved the clinical outcomes of distal radius fractures [12,13,14]. These plates provide stable fixation and have been associated with better functional recovery and lower complication rates compared to traditional fixation methods. Despite these advancements, unresolved challenges remain, particularly regarding the fixation of the volar lunate facet fragment. The volar lunate fossa, a concave depression on the volar aspect of the distal radius, plays a critical role in wrist stability. Inadequate fixation of this fragment can lead to volar subluxation of the carpal bones, compromising wrist function and leading to poor clinical outcomes [15,16,17,18].

To address these challenges, it is essential to understand the normal morphology of the volar lunate facet and its individual variations. Such knowledge is invaluable for preoperative planning and intraoperative decision-making, ensuring accurate reduction and fixation during osteosynthesis of distal radius fractures. By analyzing the 3D morphology of healthy individuals, surgeons can develop a reference framework that aids in identifying deviations in fracture cases. Previously, our research has highlighted significant gender differences in the 3D morphology of the distal radius [19]. We assessed the standard bone shapes separately for males and females, revealing notable differences in the positional relationships of key reference points, such as the tip of the radial styloid process and the volar−dorsal edges of the sigmoid notch. These findings suggest that gender-specific considerations may be necessary when planning surgical interventions for distal radius fractures. In particular, the morphology of the lunate facet is clinically varied, and accurate anatomical knowledge is considered essential for accurate repositioning and fixation of this site.

The primary objective of this study was to elucidate the relationship between the morphology of the volar lunate facet and the transverse diameter of the distal radius, with a particular focus on gender differences. We hypothesized that there are significant gender differences in the morphology of the volar lunate facet and that these differences are correlated with the transverse diameter of the distal radius. By defining 3D reference points in a normal wrist, we aimed to establish standard bone morphology for the volar lunate facet to serve as a guide for surgical interventions.

## 2. Methods

The study protocol received approval from the Institutional Review Board (Approval Code: T2019-0178). This retrospective case-control study (level of evidence III) utilized a radiographic database to identify patients who had undergone CT scans of their normal wrists. We analyzed CT images of unaffected wrists, which were taken for comparison with the side affected by distal radius fracture. Confirmation of no prior history or complaints related to the unaffected wrists was obtained through interviews and medical records. The study included CT-imaging data from 64 patients (30 males, 34 females) collected at a university hospital between January 2016 and December 2020. The average age was 50.2 ± 19.0 years for males (range 20–87) and 56.0 ± 16.1 years for females (range 23–78). Patients were excluded if they had a history of traumatic arm injuries or were under 18 years of age. CT scans were performed using a tube setting of 120 kV and 100 mAS, with a section thickness of 1–1.5 mm and a pixel size of 0.3 × 0.3 mm^2^ (Sensation Cardiac, Siemens, Tokyo, Japan). The imaging spanned from the metacarpal bone level to approximately 13 cm proximal to the radial joint surface.

### 2.1. Bone Morphology Analysis in the 3D Image

The present study’s 3D analysis encompassed creating a 3D model, establishing a coordinate system for the model, and performing analysis using reference points, as outlined in our prior research [19]. We utilized computer-based analysis software (Zed-Trauma distal radius stage by LEXI Co., Ltd., Tokyo, Japan, and BoneSimulator by Orthree, Osaka, Japan) for the analysis of the distal radius using the 3D bone model. The DICOM dataset from the CT scan served as the basis for data analysis. After the image data had been imported into the software, the radius bone was segmented according to CT values. A surface-construction algorithm was employed to build a 3D surface model. For advanced analysis, we utilized the standard triangulated (STL) data from the 3D bone model. As detailed in a previous study, we employed the data-measurement mode in BoneSimulator to establish the coordinate system based on 3D data from the distal radius.

The long axis of the radius was identified automatically through a series of steps. Initially, the software identified the central curve of the radius shaft from the proximal to the distal end by examining cross-sections at various levels. Subsequently, it calculated the central point at each level using surface data from the radial diaphysis. Finally, the long axis of the radius was defined as a straight line connecting these central points. In the 3D coordinate system, the long axis of the radius was designated as the y-axis (positive in the proximal direction, negative in the distal direction). The z-axis (positive in the radial direction, negative in the ulnar direction) was aligned with the orthogonal projection of a line originating at the base of the sigmoid notch of the distal radius and extending to the radial styloid process on a plane perpendicular to the y-axis. The x-axis (positive in the volar direction, negative in the dorsal direction) was defined as perpendicular to the y−z plane. The coronal, sagittal, and axial planes corresponded to the y−z, x−y, and x−z planes, respectively. The origin of the coordinate axes was established at the intersection of the articular surface and the long axis.

Figure 1 shows the measured reference points and the measurement parameters. In the sagittal plane, along the radial bone axis, a line parallel to the volar cortex of the diaphysis was configured. The starting point of the inclination towards the volar margin of the distal end was defined as reference point 1; the volar edge of the joint on the bone axis was defined as reference point 2; and the volar edge of the lunate facet at the distal end of the radius was defined as reference point 3 (Figure 1). From the 3D coordinates of each reference point, the bone-axis distance, volar−dorsal distance, radial−ulnar distance, and 3D straight-line distance were measured (Figure 2a–d). Additionally, the angle formed between the volar cortex of the diaphysis and the line connecting reference points 1−2 and 1−3 in the sagittal plane (inclination angle, Figure 2e) and the angle formed between the line connecting reference points 2−3 and the z-axis in the axial plane (inclination angle, Figure 2f) were measured. In the coronal plane, the maximum transverse diameter of the distal radius was measured from the most radial end to the most ulnar end perpendicular to the radial long axis (Figure 1).

### 2.2. Statistical Analysis

Results are presented as the mean ± standard deviation. We analyzed the average positions of the three reference points relative to the origin for each point. The Shapiro−Wilk test was used to assess the normality of the datasets. Gender differences in the parameters for each reference point were compared, along with morphometric indices of the distal radius. Welch’s *t*-test was employed to compare male and female measurements. Significance was defined as *p* values less than 0.05. Pearson correlation analysis (Microsoft^®^ Excel365) was used to evaluate the relationship between the morphometric indices of the distal radius and the transverse diameter.

## 3. Results

Table 1, Table 2 and Table 3 show the results of the measurements between the reference points. The transverse diameter was 30.0 ± 2.8 mm (males: 32.0 ± 2.6 mm, females: 28.3 ± 1.6 mm, *p* < 0.05 between males and females). In the measurement between reference points 1−3 and 2−3, there were significant gender differences in the volar−dorsal distance, 3D straight-line distance, and inclination angle (*p* < 0.05). Additionally, there was a significant gender difference in the radial−ulnar distance in the measurement between reference points 1−3 (*p* < 0.05). The inclination angles were significantly larger in the males compared to the females. This suggests that males have steeper angles for the volar surface.

Table 4, Table 5 and Table 6 show the correlations between transverse diameter and each parameter. In the measurements between reference points 1−3 and 2−3; the volar−dorsal distance; radial−ulnar distance; and 3D straight-line distance showed significant correlations with the transverse diameter of the radius. This suggests that the larger the transverse diameter; the longer the radial−ulnar distance of the lunate facet; the greater the volar−dorsal distance; and the greater the distance from the inclined base.

## 4. Discussion

The results of this study indicate that males have larger transverse diameters, greater protrusion of the volar lunate facet, greater distances from the bone axis to the protrusion of the volar lunate facet, and larger inclination angles compared to females. Additionally, a larger transverse diameter is associated with greater protrusion of the volar lunate facet, longer distances from the bone axis to the protrusion of the volar lunate facet, and greater straight-line distances from the base of the inclination to the protrusion of the volar lunate facet.

Several studies have previously performed morphometric analysis of the volar cortex of the distal radius using plain X-rays and 3DCT [20,21,22,23,24,25]. All these studies have consistently shown that males have a larger transverse diameter and a steeper inclination compared to females. The results of the present study are consistent with these previous findings. Volar locking plates are considered to have an anatomical design and are widely used to restore normal alignment following distal radius fractures. However, a significant discrepancy has been demonstrated between the design of the plates and the anatomical structure of the volar surface of the distal radius. This discrepancy may be attributable to variations in the morphometry of the volar aspect.

In this study, we found significant variability in the morphology of the volar distal radius, a variability that correlates with its transverse diameter. Building on these findings, it is critical for orthopedic surgeons to understand these variations when selecting and positioning volar locking plates during surgical interventions. Improperly fitting plates can lead to complications such as malalignment, suboptimal fixation, and subsequent functional impairment. Therefore, preoperative planning should incorporate detailed morphometric assessments to ensure the selection of appropriately sized and shaped implants.

In distal radius fractures, the fixation of the volar lunate facet fragment is crucial to prevent volar subluxation of the carpal bones and to avoid wrist dysfunction [26,27,28]. Adequate coverage of the fragment is required for proper fixation [29,30]. However, as shown in this study and in previous studies, there are significant differences in bone morphology between males and females on the volar surface of the distal radius. Therefore, standard plates may not fit the bone properly, and the selection of rim plates that cover the volar edge of the joint or additional fixation may be necessary [30,31,32]. Although predicting these needs preoperatively can sometimes be challenging, the results of this study demonstrate a correlation between the transverse diameter of the radius and the protrusion of the volar edge. This has important implications because preoperative morphometric measurements may help predict the need for rim plates or additional fixations.

### Limitations

The limitations of this study include the measurements being performed by a single examiner and the small sample size. Additionally, it has been suggested that the morphology of the distal radius changes with age. The volar cortical angle decreased with age in both sexes, with men having a greater angle than women [33]. This study did not take age-related changes in bone morphology into account, which is another limitation. Future studies should include larger, more diverse populations and consider the effects of aging on bone morphology to provide a more comprehensive understanding of the anatomical variations. Another limitation is the reliance on imaging modalities that may not capture the full extent of the anatomical variations. While plain X-rays and 3DCT provide valuable insights, they may not fully represent the complex three-dimensional structures of the bone. Incorporating more advanced imaging techniques, such as MRI or high-resolution CT, could yield more detailed and accurate morphometric data. Additionally, biomechanical studies are needed to assess how these anatomical variations influence the performance of different implant designs under physiological loads. This study has not shown an improvement in clinical outcomes. Despite these limitations, the study highlights the variability in the bone morphology of the volar surface of the distal radius in relation to its transverse diameter. Based on these findings, it may be feasible to treat patients with distinctive lunate-facet geometry using specific implants or fixation techniques.

## 5. Conclusions

In conclusion, the larger the transverse diameter of the radius, the larger the protrusion of the volar surface of the distal radius lunate facet. In males, the inclination is steeper and the volar locking plate may not fit properly with the volar cortical bone of the lunate facet. These morphological characteristics should be considered when applying a volar locking plate in distal radius fractures. The measurement results presented in this study should be considered as a reference for future implant design. These measurements can provide critical data on the specific dimensions of and variations in the morphology of the distal radius, which can guide the development of implants that more closely match the anatomical contours of individual patients. This tailored approach to implant design and selection, based on thorough preoperative analysis, could enhance surgical precision, reduce the likelihood of complications such as malalignment or inadequate fixation, and ultimately lead to better clinical outcomes in the treatment of distal radius fractures. This study underscores the importance of personalized medicine and the potential benefits of integrating advanced imaging and customization technologies in orthopedic surgery.

By addressing the anatomical variations between individuals, particularly the differences between males and females, surgeons can enhance the efficacy of surgical interventions and reduce the risk of postoperative complications. The findings of this study contribute to the growing body of evidence supporting the need for personalized approaches in orthopedic-implant design and highlight the potential for future advancements in this field.

## Figures and Tables

**Figure 1 diagnostics-14-01802-f001:**
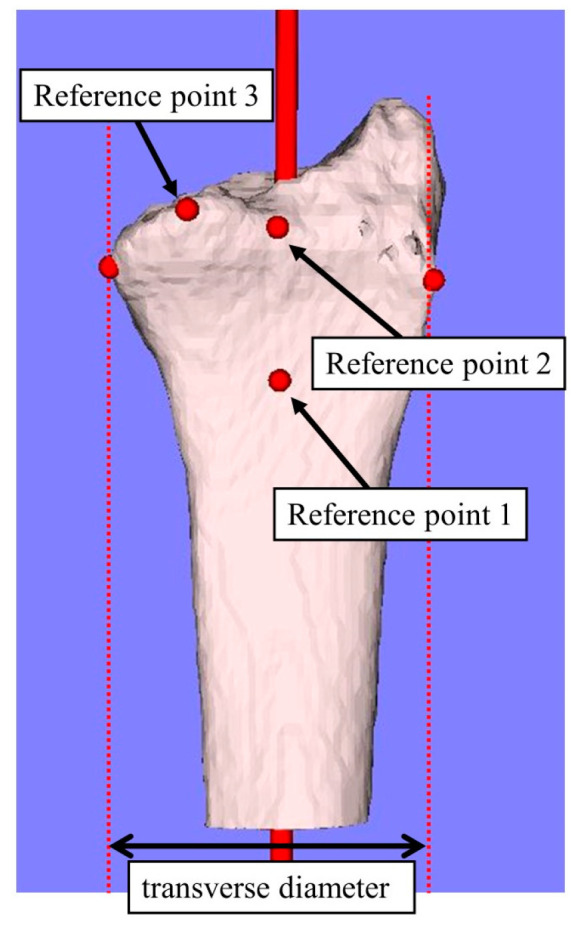
Measured reference points and the measurement parameters. The starting point of the inclination towards the volar margin of the distal end was defined as reference point 1; the volar edge of the joint on the bone axis was defined as reference point 2; and the volar edge of the lunate facet at the distal end of the radius was defined as reference point 3.

**Figure 2 diagnostics-14-01802-f002:**
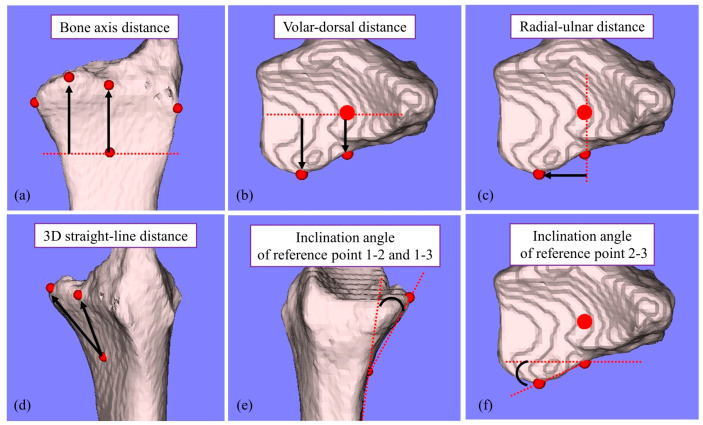
Measurement parameters. (**a**) Bone-axis distance, (**b**) Volar−dorsal distance, (**c**) Radial−ulnar distance, (**d**) 3D straight-line distance (**e**) Inclination angle A: the angle formed between the line connecting reference points 1 and 2 and the line connecting reference points 1 and 3 in the sagittal plane, (**f**) Inclination angle B: the angle formed between the line connecting reference points 2−3 and the z-axis in the axial plane.

**Table 1 diagnostics-14-01802-t001:** The results of the measurements between reference points 1 and 2.

Parameter	Measurement	Males	Females	*p* Value
Bone-Axis Distance	14.7 ± 2.9 mm	14.7 ± 2.7 mm	14.8 ± 3.1 mm	0.86
Volar−Dorsal Distance	5.5 ± 1.2 mm	5.7 ± 1.1 mm	5.4 ± 1.3 mm	0.22
Radial−Ulnar Distance	-	-	-	-
3D Straight-Line Distance	15.8 ± 2.9 mm	15.8 ± 2.7 mm	15.8 ± 3.1 mm	0.98
Inclination Angle	23.3 ± 4.6°	24.5 ± 4.7°	22.3 ± 4.3°	0.05

**Table 2 diagnostics-14-01802-t002:** The results of the measurements between reference points 1 and 3.

Parameter	Measurement	Males	Females	*p* Value
Bone-Axis Distance	15.7 ± 2.9 mm	16.0 ± 3.0 mm	15.5 ± 2.8 mm	0.53
Volar−Dorsal Distance	8.3 ± 1.5 mm	9.0 ± 1.3 mm	7.6 ± 1.4 mm	<0.05
Radial−Ulnar Distance	8.9 ± 1.5 mm	9.8 ± 1.3 mm	8.1 ± 1.2 mm	<0.05
3D Straight-Line Distance	20.0 ± 2.8 mm	20.9 ± 2.8 mm	19.1 ± 2.6 mm	<0.05
Inclination Angle	30.5 ± 5.3°	32.7 ± 5.3°	28.5 ± 4.6°	<0.05

**Table 3 diagnostics-14-01802-t003:** The results of the measurements between reference points 2 and 3.

Parameter	Measurement	Males	Females	*p* Value
Bone-Axis Distance	1.3 ± 0.9 mm	1.5 ± 0.9 mm	1.1 ± 0.9 mm	0.06
Volar−Dorsal Distance	2.7 ± 1.0 mm	3.3 ± 0.9 mm	2.2 ± 0.7 mm	<0.05
Radial−Ulnar Distance	8.9 ± 1.5 mm	9.8 ± 1.3 mm	8.1 ± 1.2 mm	-
3D Straight-Line Distance	9.8 ± 1.6 mm	10.5 ± 1.3 mm	8.5 ± 1.2 mm	<0.05
Inclination Angle	17.1 ± 5.1°	18.7 ± 5.2°	15.6 ± 4.7°	<0.05

**Table 4 diagnostics-14-01802-t004:** The correlations between transverse diameter and each parameter between reference points 1 and 2.

Parameter	The Correlation Coefficients
Bone-Axis Distance	0.21
Volar−Dorsal Distance	0.33
Radial−Ulnar Distance	-
3D Straight-Line Distance	0.24
Inclination Angle	0.15

**Table 5 diagnostics-14-01802-t005:** The correlations between transverse diameter and each parameter between reference points 1 and 3.

Parameter	The Correlation Coefficients
Bone-Axis Distance	0.34
Volar−Dorsal Distance	0.52
Radial−Ulnar Distance	0.69
3D Straight-Line Distance	0.55
Inclination Angle	0.16

The red values indicate significant correlations.

**Table 6 diagnostics-14-01802-t006:** The correlations between transverse diameter and each parameter between reference points 2 and 3.

Parameter	The Correlation Coefficients
Bone Axis Distance	0.28
Volar−Dorsal Distance	0.41
Radial−Ulnar Distance	0.69
3D Straight-Line Distance	0.72
Inclination Angle	0.09

The red values indicate significant correlations.

## Data Availability

The datasets analyzed during the present study are available from the corresponding author upon reasonable request.

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
