# Peer review of "Three-Dimensional Morphometric Analysis of the Volar Cortical Shape of the Lunate Facet of the Distal Radius"

_diagnostics, 2024, doi:10.3390/diagnostics14161802_

Round 1

Reviewer 1 Report

Comments and Suggestions for Authors

The authors perform an anatomic study of the distal radius comparing gender. They find that there are significant differences in some of the morphometric measurements of the distal radius.

The study may be a good preliminary study when designing a prosthesis. The authors should better explain the reason for performing the study and how this could have a clinical impact.

Author Response

The authors perform an anatomic study of the distal radius comparing gender. They find that there are significant differences in some of the morphometric measurements of the distal radius.

The study may be a good preliminary study when designing a prosthesis. The authors should better explain the reason for performing the study and how this could have a clinical impact.

Response)Thank you for reviewing our manuscript. We have added the descriptions for the reason and clinical impact. (Line 75-77, Line 262-268)

Reviewer 2 Report

Comments and Suggestions for Authors

Discussion

Strengths:

  • The discussion effectively interprets the study's findings in the context of the broader literature on distal radius morphology.
  • The section clearly explains the clinical implications of the research, highlighting the importance of personalized approaches in orthopedic surgery.
  • The limitations of the study are acknowledged and suggestions are made for how future research could address these limitations.

Weaknesses & Suggestions:

  • Transitions: The discussion could benefit from more transitional phrases to guide the reader through the progression of ideas. For example, "Building on these findings...", "This has important implications because...", or "Despite these limitations, the study suggests...".
  • Conciseness: Some sentences are wordy and could be condensed for clarity. For example, "The volar cortical angle decreased with age in both men and women, and men had a greater volar cortical angle than women [26]." could become "The volar cortical angle decreased with age in both sexes, with men having a greater angle than women [26]."
  • Clarity: A few sections are unclear. For example, "The measurement results presented in this study should be considered as a reference for future implant design." - could you expand on how exactly the measurements could inform design?
  • Organization: The discussion jumps a bit between highlighting the importance of personalized medicine, acknowledging limitations, and discussing the need for future research. Consider breaking this into clear subsections or using subheadings to organize these different ideas.
Comments on the Quality of English Language

Discussion

Strengths:

  • The discussion effectively interprets the study's findings in the context of the broader literature on distal radius morphology.
  • The section clearly explains the clinical implications of the research, highlighting the importance of personalized approaches in orthopedic surgery.
  • The limitations of the study are acknowledged and suggestions are made for how future research could address these limitations.

Weaknesses & Suggestions:

  • Transitions: The discussion could benefit from more transitional phrases to guide the reader through the progression of ideas. For example, "Building on these findings...", "This has important implications because...", or "Despite these limitations, the study suggests...".
  • Conciseness: Some sentences are wordy and could be condensed for clarity. For example, "The volar cortical angle decreased with age in both men and women, and men had a greater volar cortical angle than women [26]." could become "The volar cortical angle decreased with age in both sexes, with men having a greater angle than women [26]."
  • Clarity: A few sections are unclear. For example, "The measurement results presented in this study should be considered as a reference for future implant design." - could you expand on how exactly the measurements could inform design?
  • Organization: The discussion jumps a bit between highlighting the importance of personalized medicine, acknowledging limitations, and discussing the need for future research. Consider breaking this into clear subsections or using subheadings to organize these different ideas.

Author Response

Strengths:

The discussion effectively interprets the study's findings in the context of the broader literature on distal radius morphology.

The section clearly explains the clinical implications of the research, highlighting the importance of personalized approaches in orthopedic surgery.

The limitations of the study are acknowledged and suggestions are made for how future research could address these limitations.

Response) Thank you for your insightful comments. We are pleased that the interpretation of the study results regarding distal radius morphology was clear and effective. We strived to ensure that our discussion not only situates our results within the existing body of research but also advances the personalized approaches in orthopedic surgery.

Weaknesses & Suggestions:

Transitions: The discussion could benefit from more transitional phrases to guide the reader through the progression of ideas. For example, "Building on these findings...", "This has important implications because...", or "Despite these limitations, the study suggests...".

Response) Thank you for suggestion. We rephrased some of the sentences. (Line 214-216, Line 241, Line 258-262)

Conciseness: Some sentences are wordy and could be condensed for clarity. For example, "The volar cortical angle decreased with age in both men and women, and men had a greater volar cortical angle than women [26]." could become "The volar cortical angle decreased with age in both sexes, with men having a greater angle than women [26]."

Response) Thank you for comment. We revised some of the sentences to simplify the claim. (Line 31-32, Line 214-216, Line 247-248)

Clarity: A few sections are unclear. For example, "The measurement results presented in this study should be considered as a reference for future implant design." - could you expand on how exactly the measurements could inform design?

Response) Thank you for the comment. We added some descriptions. (Line 269-275)

Organization: The discussion jumps a bit between highlighting the importance of personalized medicine, acknowledging limitations, and discussing the need for future research. Consider breaking this into clear subsections or using subheadings to organize these different ideas.

Response) Thank you for the comment. We agree that the importance of personalized medicine is a bit out of the focus of the paper. This section was removed and a sub-heading on limitations was added. (Line 244)

Reviewer 3 Report

Comments and Suggestions for Authors

Thank you for the opportunity to review this interesting and well-written article.

Here are some suggestions to improve the manuscript:

INTRODUCTION

  • line 31-38: some of these affirmations would need some bibliography

METHODS

  • line 78: this sentence is not clear… Unaffected from what?
  • line 139-142: are those lines supposed to be in the Figure 1 description or in the text?

RESULTS

ok

DISCUSSION (not discussionS)

  • Line 216-221: do you think this would improve clinical outcomes? Has it been proven?
  • Line 236-252: make another paragraph called conclusion

Author Response

Thank you for the opportunity to review this interesting and well-written article.

Here are some suggestions to improve the manuscript:

INTRODUCTION

line 31-38: some of these affirmations would need some bibliography

Response)We have added some references in this paragraph.

METHODS

line 78: this sentence is not clear… Unaffected from what?

Response) We revised the description. (Line 90)

line 139-142: are those lines supposed to be in the Figure 1 description or in the text?

Response)These lines are the legend for Figure 1. We have changed the font size so that can see that these lines are legends. (Line 157-159)

RESULTS

ok

Response)Thank you.

DISCUSSION (not discussionS)

Line 216-221: do you think this would improve clinical outcomes? Has it been proven?

Response)Improved clinical outcomes have not yet been demonstrated. Based on this finding, it may be feasible to treat patients with a distinctive lunate facet geometry with specific implants or specific fixation techniques. We added the descriptions. (Line 258-262)

Line 236-252: make another paragraph called conclusion

Response)We made another paragraph for the conclusion. (Line 263-325)